# A Universal Approximation Theorem of Deep Neural Networks for Expressing Probability Distributions

**Yulong Lu**
Department of Mathematics and Statistics
University of Massachusetts Amherst
Amherst, MA 01003
lu@math.umass.edu

**Jianfeng Lu**
Mathematics Department
Duke University
Durham, NC 27708
jianfeng@math.duke.edu

## Abstract

This paper studies the universal approximation property of deep neural networks for representing probability distributions. Given a target distribution $\pi$ and a source distribution $p_z$ both defined on $\mathbb{R}^d$, we prove under some assumptions that there exists a deep neural network $g : \mathbb{R}^d \to \mathbb{R}$ with ReLU activation such that the push-forward measure $(\nabla g)_\# p_z$ of $p_z$ under the map $\nabla g$ is arbitrarily close to the target measure $\pi$. The closeness are measured by three classes of integral probability metrics between probability distributions: 1-Wasserstein distance, maximum mean distance (MMD) and kernelized Stein discrepancy (KSD). We prove upper bounds for the size (width and depth) of the deep neural network in terms of the dimension $d$ and the approximation error $\varepsilon$ with respect to the three discrepancies. In particular, the size of neural network can grow exponentially in $d$ when 1-Wasserstein distance is used as the discrepancy, whereas for both MMD and KSD the size of neural network only depends on $d$ at most polynomially. Our proof relies on convergence estimates of empirical measures under aforementioned discrepancies and semi-discrete optimal transport.

## 1  Introduction

In recent years, deep learning has achieved unprecedented success in numerous machine learning problems [29, 51]. The success of deep learning is largely attributed to the usage of deep neural networks (DNNs) for representing and learning the unknown structures in machine learning tasks, which are usually modeled by some unknown function mappings or unknown probability distributions. The effectiveness of using neural networks (NNs) in approximating functions has been justified rigorously in the last three decades. Specifically, a series of early works [12, 17, 24, 7] on universal approximation theorems show that a continuous function defined on a bounded domain can be approximated by a sufficiently large shallow (two-layer) neural network. In particular, the result by [7] quantifies the approximation error of shallow neural networks in terms of the decay property of the Fourier transform of the function of interest. Recently, the expressive power of DNNs for approximating functions have received increasing attention starting from the works by [34] and [56]; see also [57, 45, 47, 41, 48, 14, 40] for more recent developments. The theoretical benefits of using deep neural networks over shallow neural networks have been demonstrated in a sequence of depth separation results; see e.g. [16, 52, 54, 13]

Compared to a vast number of theoretical results on neural networks for approximating functions, the use of neural networks for expressing distributions is far less understood on the theoretical side. The idea of using neural networks for modeling distributions underpins an important class of unsupervised learning techniques called *generative models*, where the goal is to approximate or learn complex probability distributions from the training samples drawn from the distributions. Typical generative

models include Variational Autoencoders [27], Normalizing Flows [46] and Generative Adversarial Networks (GANs) [18], just to name a few. In these generative models, the probability distribution of interest can be very complex or computationally intractable, and is usually modelled by transforming a simple distribution using some map parameterized by a (deep) neural network. In particular, a GAN consists of a game between a generator and a discriminator which are represented by deep neural networks: the generator attempts to generate fake samples whose distribution is indistinguishable from the real distribution and it generate samples by mapping samples from a simple input distribution (e.g. Gaussian) via a deep neural network; the discriminator attempts to learn how to tell the fake apart from the real. Despite the great empirical success of GANs in various applications, its theoretical analysis is far from complete. Existing theoretical works on GANs are mainly focused on the trade-off between the generator and the discriminator (see e.g. [39, 2, 3, 35, 5]). The key message from these works is that the discriminator family needs to be chosen appropriately according to the generator family in order to obtain a good generalization error.

**Our contributions.** In this work, we focus on an even more fundamental question on GANs and other generative models which is not yet fully addressed. Namely how well can DNNs express probability distributions? We shall answer this question by making the following contributions.

• Given a fairly general source distribution and a target distribution defined on $\mathbb{R}^d$ which satisfies certain integrability assumptions, we show that there is a ReLU DNN with $d$ inputs and one output such that the push-forward of the source distribution via the gradient of the output function defined by the DNN is arbitrarily close to the target. We measure the closeness between probability distributions by three integral probability metrics (IPMs): 1-Wasserstein metric, maximum mean discrepancy and kernelized Stein discrepancy.
• Given a desired approximation error $\varepsilon$, we prove complexity upper bounds for the depth and width of the DNN needed to attain the given approximation error with respect to the three IPMs mentioned above; our complexity upper bounds are given with explicit dependence on the dimension $d$ of the target distribution and the approximation error $\varepsilon$.

It is also worth mentioning that the DNN constructed in the paper is explicit: the output function of the DNN is the maximum of finitely many (multivariate) affine functions, with the affine parameters determined explicitly in terms of the source measure and target measure.

**Related work.** Let us discuss some previous related work and compare the results there with ours. Kong and Chaudhuri [28] analyzed the expressive power of normalizing flow models under the $L^1$-norm of distributions and showed that the flow models have limited approximation capability in high dimensions. We show that feedforward DNNs can approximate a general class of distributions in high dimensions with respect to three IPMs. Lin and Jegelka [36] studied the universal approximation of certain ResNets for multivariate functions; the approximation result there however was not quantitative and did not consider the universal approximation of ResNets for distributions. The work by Bailey and Telgarsky [6] considered the approximation power of DNNs for expressing uniform and Gaussian distributions in the Wasserstein distance, whereas we prove quantitative approximation results for fairly general distributions under three IPMs. The work by Lee et al. [30] is closest to ous, where the authors considered a class of probability distributions that are given as push-forwards of a base distribution by a class of Barron functions, and showed that those distributions can be approximated in Wasserstein metrics by push-forwards of NNs, essentially relying on the ability of NNs for approximating Barron functions. It is however not clear what probability distributions are given by push-forward of a base one by Barron functions. In this work, we provide more explicit and direct criteria of the target distributions.

**Notations.** Let us introduce several definitions and notations to be used throughout the paper. We start with the definition of a fully connected and feed-forward neural network.

**Definition 1.1.** *A (fully connected and feed-forward) neural network of $L$ hidden layers takes an input vector $x \in \mathbb{R}^{N_0}$, outputs a vector $y \in \mathbb{R}^{N_{L+1}}$ and has $L$ hidden layers of sizes $N_1, N_2, \cdots N_L$. The neural network is parametrized by the weight matrices $W^\ell \in \mathbb{R}^{N_{\ell-1} \times N_\ell}$ and bias vectors $b^\ell$ with $\ell = 1, 2, \cdots, L+1$. The output $y$ is defined from the input $x$ iteratively according to the following.*

$$
\begin{aligned}
x^0 &= x, \\
x^\ell &= \sigma(W^{\ell-1}x^{\ell-1} + b^{\ell-1}), \ 1 \le \ell \le L \\
y &= W^L x^L + b^L.
\end{aligned}
\tag{1.1}
$$

*Here $\sigma$ is a (nonlinear) activation function which acts on a vector $x$ component-wisely, i.e. $[\sigma(x)]_i = \sigma(x_i)$. When $N_1 = N_2 = \cdots = N_L = N$, we say the network network has width $N$ and depth $L$. The neural network is said to be a deep neural network (DNN) if $L \geq 2$. The function defined by the deep neural network is denoted by $\mathrm{DNN}(\{W^\ell, b^\ell\}_{\ell=1}^{L+1})$.*

Popular choices of activation functions $\sigma$ include the rectified linear unit (ReLU) function $\mathrm{ReLU}(x) = \max(x, 0)$ and the sigmoid function $\mathrm{Sigmoid}(x) = (1 + e^{-x})^{-1}$.

Given a matrix $A$, let us denote its $n$-fold direct sum by $\oplus^n A = \overbrace{A \oplus A \oplus \cdots \oplus A}^{n \text{ times}} = \mathrm{diag}(A, \cdots, A)$. We denote by $\mathcal{P}_2(\mathbb{R}^d)$ the space of probability measures with finite second moment. Given two probability measures $\mu$ and $\nu$ on $\mathbb{R}^d$, a transport map $T$ between $\mu$ and $\nu$ is a measurable map $T : \mathbb{R}^d \to \mathbb{R}^d$ such that $\nu = T_\# \mu$ where $T_\# \mu$ denotes the push-forward of $\mu$ under the map $T$, i.e., for any measurable $A \subset \mathbb{R}^d$, $\nu(A) = \mu(T^{-1}(A))$. We denote by $\Gamma(\mu, \nu)$ the set of transport plans between $\mu$ and $\nu$ which consists of all coupling measures $\gamma$ of $\mu$ and $\nu$, i.e., $\gamma(A \times \mathbb{R}^d) = \mu(A)$ and $\gamma(\mathbb{R}^d \times B) = \nu(B)$ for any measurable $A, B \subset \mathbb{R}^d$. We may use $C, C_1, C_2$ to denote generic constants which do not depend on any quantities of interest (e.g. dimension $d$).

The rest of the paper is organized as follows. We describe the problem and state the main result in Section 2. Section 3 and Section 4 devote to the two ingredients for proving the main result: convergence of empirical measures in IPMs and building neural-network-based maps between the source measure and empirical measures via semi-discrete optimal transport respectively. Proofs of lemmas and intermediate results are provided in appendices.

## 2   Problem Description and Main Result

Let $\pi(x)$ be the target probability distribution defined on $\mathbb{R}^d$ which one would like to learn or generate samples from. In the framework of GANs, one is interested in representing the distribution $\pi$ implicitly by a generative neural network. Specifically, let $\mathcal{G}_{\mathrm{NN}} \subset \{g : \mathbb{R}^d \to \mathbb{R}^d\}$ be a subset of generators (transformations), which are defined by neural networks. The concrete form of $\mathcal{G}_{\mathrm{NN}}$ is to be specified later. Let $p_z$ be a source distribution (e.g. standard normal). The push-forward of $p_z$ under the transformation $g \in \mathcal{G}_{\mathrm{NN}}$ is denoted by $p_x = g_\# p_z$. In a GAN problem, one aims to find $g \in \mathcal{G}_{\mathrm{NN}}$ such that $g_\# p_z \approx \pi$. In the mathematical language, GANs can be formulated as the following minimization problem:

$$\inf_{g \in \mathcal{G}_{\mathrm{NN}}} D(g_\# p_z, \pi) \tag{2.1}$$

where $D(p, \pi)$ is some discrepancy measure between probability measures $p$ and $\pi$, which typically takes the form of integral probability metric (IPM) or adversarial loss defined by

$$D(p, \pi) = d_{\mathcal{F}_D}(p, \pi) := \sup_{f \in \mathcal{F}_D} \left| \mathbf{E}_{X \sim p} f(X) - \mathbf{E}_{X \sim \pi} f(X) \right|, \tag{2.2}$$

where $\mathcal{F}_D$ is certain class of test (or witness) functions. As a consequence, GANs can be formulated as the minimax problem

$$\inf_{g \in \mathcal{G}_{\mathrm{NN}}} \sup_{f \in \mathcal{F}_D} \left| \mathbf{E}_{X \sim p} f(X) - \mathbf{E}_{X \sim \pi} f(X) \right|.$$

The present paper aims to answer the following fundamental questions on GANs:

(1) Is there a neural-network-based generator $g \in \mathcal{G}_{\mathrm{NN}}$ such that $D(g_\# p_z, \pi) \approx 0$?

(2) How to quantify the complexity (e.g. depth and width) of the neural network?

As we shall see below, the answers to the questions above depend on the IPM $D$ used to measure the discrepancy between distributions. In this paper, we are interested in three IPMs which are commonly used in GANs, including 1-Wasserstein distance [55, 1], maximum mean discrepancy [20, 15, 33] and kernelized Stein discrepancy [37, 11, 25].

**Wasserstein Distance**: When the witness class $\mathcal{F}_D$ is chosen as the the class of 1-Lipschitz functions, i.e. $\mathcal{F}_D := \{f : \mathbb{R}^d \to \mathbb{R} : \mathrm{Lip}\,(f) \leq 1\}$, the resulting IPM $d_{\mathcal{F}_D}$ becomes the 1-Wasserstein distance (also known as Kantorovich-Rubinstein distance):

$$\mathcal{W}_1(p, \pi) = \inf_{\gamma \in \Gamma(p, \pi)} \int |x - y| \gamma(dxdy).$$

The Wasserstein-GAN proposed by [1] leverages the Wasserstein distance as the objective function to improve the stability of training of the original GAN based on the Jensen-Shannon divergence. Nevertheless, it has been shown that Wasserstein-GAN still suffers from the mode collapse issue [22] and does not generalize with any polynomial number of training samples [2].

**Maximum Mean Discrepancy (MMD)**: When $\mathcal{F}_D$ is the unit ball of a reproducing kernel Hilbert space (RKHS) $\mathcal{H}_k$, i.e. $\mathcal{F}_D := \{f \in \mathcal{H}_k : \|f\|_{\mathcal{H}_k} \leq 1\}$, the resulting IPM $d_{\mathcal{F}_D}$ coincides with the maximum mean discrepancy (MMD) [20]:

$$\mathrm{MMD}(p, \pi) = \sup_{\|f\|_{\mathcal{H}_k} \leq 1} \left| \mathbf{E}_{X \sim p} f(X) - \mathbf{E}_{X \sim \pi} f(X) \right|.$$

GANs based on minimizing MMD as the loss function were firstly proposed in [15, 33]. Since MMD is a weaker metric than 1-Wasserstein distance, MMD-GANs also suffer from the mode collapse issue, but empirical results (see e.g. [8]) suggest that they require smaller discriminative networks and hence enable faster training than Wasserstein-GANs.

**Kernelized Stein Discrepancy (KSD)**: If the witness class $\mathcal{F}_D$ is chosen as $\mathcal{F}_D := \{\mathcal{T}_\pi f : f \in \mathcal{H}_k$ and $\|f\|_{\mathcal{H}_k} \leq 1\}$, where $\mathcal{T}_\pi$ is the Stein-operator defined by

$$\mathcal{T}_\pi f := \nabla \log \pi \cdot f + \nabla \cdot f, \tag{2.3}$$

the associated IPM $d_{\mathcal{F}_D}$ becomes the Kernelized Stein Discrepancy (KSD) [37, 11]:

$$\mathrm{KSD}(p, \pi) = \sup_{\|f\|_{\mathcal{H}_k} \leq 1} \mathbf{E}_{X \sim p}[\mathcal{T}_\pi f(X)].$$

The KSD has received great popularity in machine learning and statistics since the quantity $\mathrm{KSD}(p, \pi)$ is very easy to compute and does not depend on the normalization constant of $\pi$, which makes it suitable for statistical computation, such as hypothesis testing [19] and statistical sampling [38, 10]. The recent paper [25] adopts the GAN formulation (2.1) with KSD as the training loss to construct a new sampling algorithm called Stein Neural Sampler.

## 2.1 Main result

Throughout the paper, we consider the following assumptions on the reproducing kernel $k$:

**Assumption K1.** *The kernel $k$ is integrally strictly positive definite: for all finite non-zero signed Borel measures $\mu$ defined on $\mathbb{R}^d$,*

$$\iint_{\mathbb{R}^d} k(x, y) d\mu(x) d\mu(y) > 0.$$

**Assumption K2.** *There exists a constant $K_0 > 0$ such that*

$$\sup_{x \in \mathbb{R}^d} |k(x, x)| \leq K_0. \tag{2.4}$$

**Assumption K3.** *The kernel function $k : \mathbb{R}^d \times \mathbb{R}^d \to \mathbb{R}$ is twice differentiable and there exists a constant $K_1 > 0$ such that*

$$\max_{m+n \leq 1} \sup_{x,y} \|\nabla_x^m \nabla_y^n k(x, y)\| \leq K_1 \text{ and } \sup_{x,y} |\operatorname{Tr}(\nabla_x \nabla_y k(x, y))| \leq K_1(1 + d). \tag{2.5}$$

According to [51, Theorem 7], Assumption K1 is necessary and sufficient for the kernel being *characteristic*, i.e., $\mathrm{MMD}(\mu, \nu) = 0$ implies $\mu = \nu$, which guarantees that MMD is a metric. In addition, thanks to [37, Proposition 3.3], KSD is a valid discrepancy measure under the Assumption K1, namely $\mathrm{KSD}(p, \pi) \geq 0$ and $\mathrm{KSD}(p, \pi) = 0$ if and only if $p = \pi$.

Assumption K2 will be used to get an error bound for $\mathrm{MMD}(P_n, \pi)$; see Theorem 3.2. Assumption K3 will be crucial for bounding $\mathrm{KSD}(P_n, \pi)$; see Theorem 3.3. Many commonly used kernel functions fulfill all three assumptions K1-K3, including for example Gaussian kernel $k(x, y) = e^{-\frac{1}{2}|x-y|^2}$ and inverse multiquadric (IMQ) kernel $k(x, y) = (c + |x - y|^2)^\beta$ with $c > 0$ and $\beta < 0$. Unfortunately, Matérn kernels [43] only satisfy Assumptions K1-K2, but not Assumption K3 since the second order derivatives of $k$ are singular on the diagonal, so the last estimate of (2.5) is violated.

In order to bound $\mathrm{KSD}(P_n, \pi)$, we need to assume further that the target measure $\pi$ satisfies the following regularity and integrability assumptions. We will use the shorthand notation $s_\pi(x) = \nabla \log \pi(x)$.

**Assumption 1** (*L*-Lipschitz). *Assume that $s_\pi(x)$ is globally Lipschitz in $\mathbb{R}^d$, i.e. there exists a constant $\tilde{L} > 0$ such that $|s_\pi(x) - s_\pi(y)| \leq \tilde{L}|x - y|$ for all $x, y \in \mathbb{R}^d$. As a result, there exists $L > 0$ such that*

$$|s_\pi(x)| \leq L(1 + |x|) \quad \text{for all } x \in \mathbb{R}^d. \tag{2.6}$$

**Assumption 2** (sub-Gaussian). *The probability measure $\pi$ is sub-Gaussian, i.e. there exist $m = (m_1, \cdots, m_d) \in \mathbb{R}^d$ and $\upsilon > 0$ such that*

$$\mathbf{E}_{X\sim\pi}[\exp(\alpha^T(X - m))] \leq \exp(|\alpha|^2 \upsilon^2/2) \quad \text{for all } \alpha \in \mathbb{R}^d.$$

*Assume further that $\max_i |m_i| \leq m^*$ for some $m^* > 0$.*

Our main result is the universal approximation theorem for expressing probability distributions.

**Theorem 2.1** (Main theorem). *Let $\pi$ and $p_z$ be the target and the source distributions respectively, both defined on $\mathbb{R}^d$. Assume that $p_z$ is absolutely continuous with respect to the Lebesgue measure. Then under certain assumptions on $\pi$ and the kernel $k$ to be specified below, it holds that for any given approximation error $\varepsilon$, there exists a positive integer $n$, and a fully connected and feed-forward deep neural network $u = \text{DNN}(\{W^\ell, b^\ell\}_{\ell=1}^{L+1})$ of depth $L = \lceil \log_2 n \rceil$ and width $N = 2^L = 2^{\lceil \log_2 n \rceil}$, with $d$ inputs and a single output and with ReLU activation such that $d_{\mathcal{F}_D}((\nabla u)_\# p_z, \pi) \leq \varepsilon$. The complexity parameter $n$ depends on the choice of the metric $d_{\mathcal{F}_D}$. Specifically,*

*1. Consider $d_{\mathcal{F}_D} = \mathcal{W}_1$. If $\pi$ satisfies that $M_3 = \mathbf{E}_{X\sim\pi}|X|^3 < \infty$, it holds that*

$$n \leq \begin{cases} \frac{C}{\varepsilon^2}, & d = 1, \\ \frac{C \log^2(\varepsilon)}{\varepsilon^2}, & d = 2, \\ \frac{C^d}{\varepsilon^d}, & d \geq 3, \end{cases}$$

*where the constant $C$ depends only on $M_3$.*

*2. Consider $d_{\mathcal{F}_D} = MMD$ with kernel $k$. If $k$ satisfies Assumption K2, then*

$$n \leq \frac{C}{\varepsilon^2}$$

*with a constant $C$ depending only on the constant $K_0$ in (2.4).*

*3. Consider $d_{\mathcal{F}_D} = KSD$ with kernel $k$. If $k$ satisfies Assumption K3 with constant $K_1$ and if $\pi$ satisfies Assumption 1 and Assumption 2 with parameters $L, m, \upsilon$, then*

$$n \leq \frac{Cd}{\varepsilon^2},$$

*where the constant $C$ depends only on $L, K_1, m^*, \upsilon$, but not on $d$.*

Theorem 2.1 states that a given probability measure $\pi$ (with certain integrability assumption) can be approximated arbitrarily well by push-forwarding a source distribution with the gradient of a potential which can be parameterized by a finite DNN. The complexity bound in the Wasserstein case suffers from the curse of dimensionality whereas this issue was eliminated in the cases of MMD and KSD . We remark that our complexity bound for the the neural network is stated in terms of the number of widths and depths, which would lead to an estimate of number of weights needed to achieve certain approximation error.

**Proof strategy.** The proof of Theorem 2.1 is given in Appendix F, which relies on two ingredients: (1) one approximates the target $\pi$ by the empirical measure $P_n$; Proposition 3.1-Proposition 3.3 give quantitative estimates w.r.t the three IPMs defined above; (2) based on the theory of (semi-discrete) optimal transport, one can build an optimal transport map of the form $T = \nabla\varphi$ which push-forwards the source distribution $p_z$ to the empirical distribution $P_n$. Moreover, the potential $\varphi$ is explicit : it is the maximum of finitely many affine functions; it is such explicit structure that enables one represents the function $\varphi$ with a finite deep neural network; see Theorem 4.1 for the precise statement.

It is interesting to remark that our strategy of proving Theorem 2.1 shares the same spirit as the one used to prove universal approximation theorems of DNNs for functions [56, 34]. Indeed, both the universal approximation theorems in those works and ours are proved by approximating the

target function or distribution with a suitable dense subset (or sieves) on the space of functions or distributions which can be parametrized by deep neural networks. Specifically, in [56, 34] where the goal is to approximate continuous functions on a compact set, the dense sieves are polynomials which can be further approximated by the output functions of DNNs, whereas in our case we use empirical measures as the sieves for approximating distributions, and we show that empirical measures are exactly expressible by transporting a source distribution with neural-network-based transport maps.

We also remark that the push-forward map between probability measures constructed in Theorem 2.1 is the gradient of a potential function given by a neural network, i.e., the neural network is used to parametrize the potential function, instead of the map itself, which is perhaps more commonly used in practice. The rational of using such gradient formulation lies in that transport maps between two probability measures are discontinuous in general. This occurs particularly when the target measure has disjoint modes (or supports) and the input has a unique mode, which is ubiquitous in GAN applications where images are concentrated on disjoint modes while the input is chosen as Gaussian. In such case it is impossible to generate the target measure using usual NNs as the resulting functions are all continuous (in fact Lipschitz for ReLU activation and smooth for Sigmoid activation). Thus, from a theoretical viewpoint, it is advantageous to use NNs to parametrize the Brenier's potential since it is more regular (at least Lipschitz by OT theory), and to use its gradient as the transport map. From a practical perspective, the idea of taking gradients of NNs has already been used and received increasing attention in practical learning problems; see [32, 23, 42, 53] for an application of such parameterization in learning optimal transport maps and improving the training of Wasserstein-GANs; see also [58] for learning interatomic potentials and forces for molecular dynamics simulations.

## 3 Convergence of Empirical Measures in Various IPMs

In this section, we consider the approximation of a given target measure $\pi$ by empirical measures. More specifically, let $\{X_i\}_{i=1}^n$ be an i.i.d. sequence of random samples from the distribution $\pi$ and let $P_n = \frac{1}{n}\sum_{i=1}^n \delta_{X_i}$ be the empirical measure associated to the samples $\{X_i\}_{i=1}^n$. Our goal is to derive quantitative error estimates of $d_{\mathcal{F}_D}(P_n, \pi)$ with respect to three IPMs $d_{\mathcal{F}_D}$ described in the last section.

We first state an upper bound on $\mathcal{W}_1(P_n, \pi)$ in the average sense in the next proposition.

**Proposition 3.1** (Convergence in 1-Wasserstein distance). *Consider the IPM with $\mathcal{F}_D = \{f : \mathbb{R}^d \to \mathbb{R} : \mathrm{Lip}(f) \leq 1\}$. Assume that $\pi$ satisfies that $M_3 = \mathbb{E}_{X \sim \pi}|X|^3 < \infty$. Then there exists a constant $C$ depending on $M_3$ such that*

$$\mathbf{E}\mathcal{W}_1(P_n, \pi) \leq C \cdot \begin{cases} n^{-1/2}, & d = 1, \\ n^{-1/2}\log n, & d = 2, \\ n^{-1/d}, & d \geq 3. \end{cases}$$

The convergence rates of $\mathcal{W}_1(P_n, \pi)$ as stated in Proposition 3.1 are well-known in the statistics literature. The statement in Proposition 3.1 is a combination of results from [9] and [31]; see Appendix A for a short proof. We remark that the prefactor constant $C$ in the estimate above can be made explicit. In fact, one can easily obtain from the moment bound in Proposition C.1 of Appendix C that if $\pi$ is sub-Gaussian with parameters $m$ and $v$, then the constant $C$ can be chosen as $C = C'\sqrt{d}$, with some constant $C'$ depending only on $v$ and $\|m\|_\infty$. Moreover, one can also obtain a high probability bound for $\mathcal{W}_1(P_n, \pi)$ if $\pi$ is sub-exponential (see e.g., [31, Corollary 5.2]). Here we content ourselves with the expectation result as it comes with weaker assumptions and also suffices for our purpose of showing the existence of an empirical measure with desired approximation rate.

Moving on to approximation in MMD, the following proposition gives a high-probability non-asymptotic error bound of $\mathrm{MMD}(P_n, \pi)$.

**Proposition 3.2** (Convergence in MMD). *Consider the IPM with $\mathcal{F}_D = \{f \in \mathcal{H}_k : \|f\|_{H_k} \leq 1\}$. Assume that the kernel $k$ satisfies Assumption K2 with constant $K_0$. Then for every $\tau > 0$, with probability at least $1 - 2e^{-\tau}$,*

$$\mathrm{MMD}(P_n, \pi) \leq 2\sqrt{\frac{\sqrt{K_0}}{n}} + 3\sqrt{\frac{2\sqrt{K_0}\tau}{n}}.$$

Proposition 3.2 can be viewed as a special case of [50, Theorem 3.3] where the kernel class is a skeleton. Since its proof is short, we provide the proof in Appendix B for completeness.

In the next proposition, we consider the convergence estimate of empirical measures $P_n$ to $\pi$ in KSD . To the best of our knowledge, this is the first estimate on empirical measure under the KSD in the literature. This result can be useful to obtain quantitative error bounds for the new GAN/sampler called Stein Neural Sampler [25]. The proof relies on a Bernstein type inequality for the distribution of von Mises' statistics; the details are deferred to Appendix C.

**Proposition 3.3** (Convergence in KSD). *Consider the IPM with $\mathcal{F}_D := \{\mathcal{T}_\pi f : f \in \mathcal{H}_k \text{ and } \|f\|_{\mathcal{H}_k} \leq 1\}$, where $\mathcal{T}_\pi$ is the Stein operator defined in (2.3). Suppose that the kernel $k$ satisfies Assumption K3 with constant $K_1$. Suppose also that $\pi$ satisfies Assumption 1 and Assumption 2. Then for any $\delta > 0$ there exists a constant $C = C(L, K_1, m^*, \delta)$ such that with probability at least $1 - \delta$,*

$$\mathrm{KSD}(P_n, \pi) \leq C\sqrt{\frac{d}{n}}. \tag{3.1}$$

**Remark 3.1.** *Proposition 3.3 provides a non-asymptotic high probability error bound for the convergence of the empirical measure $P_n$ converges to $\pi$ in KSD. Our result implies in particular that $\mathrm{KSD}(P_n, \pi) \to 0$ with the asymptotic rate $\mathcal{O}(\sqrt{\frac{d}{n}})$. We also remark that the rate $\mathcal{O}(n^{-1/2})$ is optimal and is consistent with the asymptotic CLT result for the corresponding U-statistics of $\mathrm{KSD}^2(P_n, \pi)$ (see [37, Theorem 4.1 (2)]).*

## 4 Constructing Transport Maps via Semi-discrete Optimal Transport

In this section, we aim to build a neural-network-based map which push-forwards a given source distribution to discrete probability measures, including in particular the empirical measures. The main result of this section is the following theorem.

**Theorem 4.1.** *Let $\mu \in \mathcal{P}_2(\mathbb{R}^d)$ be absolutely continuous with respect to the Lebesgue measure with Radon–Nikodym density $\rho(x)$. Let $\nu = \sum_{i=1}^n \nu_i \delta_{y_i}$ for some $\{y_j\}_{j=1}^n \subset \mathbb{R}^d, \nu_j \geq 0$ and $\sum_{j=1}^n \nu_j = 1$. Then there exists a transport map of the form $T = \nabla u$ such that $T_{\#}\mu = \nu$ where $u$ is a fully connected deep neural network of depth $L = \lceil \log_2 n \rceil$ and width $N = 2^L = 2^{\lceil \log_2 n \rceil}$, and with ReLU activation function and parameters $\{W^\ell, b^\ell\}_{\ell=1}^{L+1}$ such that $u = \mathrm{DNN}(\{W^\ell, b^\ell\}_{\ell=1}^{L+1})$.*

As shown below, the transport map in Theorem 4.1 is chosen as the optimal transport map from the continuous distribution $\mu$ to the discrete distribution $\nu$, which turns out to be the gradient of a piece-wise linear function, which in turn can be expressed by neural networks. We remark that the weights and biases of the constructed neural network can also be characterized explicitly in terms of $\mu$ and $\nu$ (see the proof of Proposition 4.1). Since semi-discrete optimal transport plays an essential role in the proof of Theorem 4.1, we first recall the set-up and some key results on optimal transport in both general and semi-discrete settings.

**Optimal transport with quadratic cost.** Let $\mu$ and $\nu$ be two probability measures on $\mathbb{R}^d$ with finite second moments. Let $c(x, y) = \frac{1}{2}|x - y|^2$ be the quadratic cost. Then Monge's [44] optimal transportation problem is to transport the probability mass between $\mu$ and $\nu$ while minimizing the quadratic cost, i.e.

$$\inf_{T:\mathbb{R}^d \to \mathbb{R}^d} \int \frac{1}{2}|x - T(x)|^2 \mu(dx) \quad \text{s.t. } T_{\#}\mu = \nu. \tag{4.1}$$

A map $T$ attaining the infimum above is called an optimal transport map. In general an optimal transport map may not exist since Monge's formulation prevents splitting the mass so that the set of transport maps may be empty. On the other hand, Kantorovich [26] relaxed the problem by considering minimizing the transportation cost over transport plans instead of the transport maps:

$$\inf_{\gamma \in \Gamma(\mu, \nu)} \mathcal{K}(\gamma) := \inf_{\gamma \in \Gamma(\mu, \nu)} \int \frac{1}{2}|x - y|^2 \gamma(dxdy). \tag{4.2}$$

A coupling $\gamma$ achieving the infimum above is called an optimal coupling. Noting that problem (4.2) above is a linear programming, Kantorovich proposed a dual formulation for (4.2):

$$\sup_{(\varphi, \psi) \in \Phi_c} \mathcal{J}(\varphi, \psi) := \sup_{(\varphi, \psi) \in \Phi_c} \int \varphi d\mu + \psi d\nu,$$

where $\Phi_c$ be the set of measurable functions $(\varphi, \psi) \in L^1(\mu) \times L^1(\nu)$ satisfying $\varphi(x) + \psi(y) \leq \frac{1}{2}|x - y|^2$. We also define the $c$-transformation $\varphi^c : \mathbb{R}^d \to \mathbb{R}$ of a function $\varphi : \mathbb{R}^d \to \mathbb{R}$ by

$$\varphi^c(y) = \inf_{x \in \mathbb{R}^d} c(x, y) - \varphi(x) = \inf_{x \in \mathbb{R}^d} \frac{1}{2}|x - y|^2 - \varphi(x).$$

Similarly, one can define $\psi^c$ associated to $\psi$. The Kantorovich's duality theorem (see e.g. [55, Theorem 5.10]) states that

$$\inf_{\gamma \in \Gamma(\mu, \nu)} \mathcal{K}(\gamma) = \sup_{(\varphi, \psi) \in \Phi_c} \mathcal{J}(\varphi, \psi) = \sup_{\varphi \in L^1(d\mu)} \mathcal{J}(\varphi, \varphi^c) = \sup_{\psi \in L^1(d\nu)} \mathcal{J}(\psi^c, \psi). \qquad (4.3)$$

Moreover, if the source measure $\mu$ is absolutely continuous with respect to the Lebesgue measure, then the optimal transport map defined in Monge's problem is given by a gradient field, which is usually referred to as the *Brenier's map* and can be characterized explicitly in terms of the solution of the dual Kantorovich problem. A precise statement is included in Theorem E.1 in Appendix E.

**Semi-discrete optimal transport.** Let us now consider the optimal transport problem in the semi-discrete setting: the source measure $\mu$ is continuous and the target measure $\nu$ is discrete. Specifically, assume that $\mu \in \mathcal{P}_2(\mathbb{R}^d)$ is absolutely continuous with respect to the Lebesgue measure, i.e. $\mu(dx) = \rho(x)dx$ for some probability density $\rho$ and $\nu$ is discrete, i.e. $\nu = \sum_{j=1}^n \nu_j \delta_{y_j}$ for some $\{y_j\}_{j=1}^n \subset \mathbb{R}^d, \nu_j \geq 0$ and $\sum_{j=1}^n \nu_j = 1$. In the semi-discrete setting, Monge's problem becomes

$$\inf_T \int \frac{1}{2}|x - T(x)|^2 \mu(dx) \text{ s.t} \int_{T^{-1}(y_j)} d\mu = \nu_j, \; j = 1, \cdots, n. \qquad (4.4)$$

In this case the action of the transport map is clear: it assigns each point $x \in \mathbb{R}^d$ to one of these $y_j$. Moreover, by taking advantage of the dicreteness of the measure $\nu$, one sees that the dual Kantorovich problem in the semi-discrete case becomes maximizing the following functional

$$\mathcal{F}(\psi) = \mathcal{F}(\psi_1, \cdots, \psi_n) = \int \inf_j \left( \frac{1}{2}|x - y_j|^2 - \psi_j \right) \rho(x)dx + \sum_{j=1}^n \psi_j \nu_j. \qquad (4.5)$$

Similar to the continuum setting, the optimal transport map of Monge's problem (4.4) can be characterized by the maximizer of $\mathcal{F}$. To see this, let us introduce an important concept of power diagram (or Laguerre diagram) [4, 49]. Given a finite set of points $\{y_j\}_{j=1}^n \subset \mathbb{R}^d$ and the scalars $\psi = \{\psi_j\}_{j=1}^n$, the power diagrams associated to the scalars $\psi$ and the points $\{y_j\}_{j=1}^n$ are the sets

$$P_j := \left\{ x \in \mathbb{R}^d \; \Big| \; \frac{1}{2}|x - y_j|^2 - \psi_j \leq \frac{1}{2}|x - y_k|^2 - \psi_k, \forall k \neq j \right\}. \qquad (4.6)$$

By grouping the points according to the power diagrams $P_j$, we have from (4.5) that

$$\mathcal{F}(\psi) = \sum_{j=1}^n \left[ \int_{P_j} \left( \frac{1}{2}|x - y_j|^2 - \psi_j \right) \rho(x)dx + \psi_j \nu_j \right]. \qquad (4.7)$$

The following theorem characterizes the optimal transport map of Monge's problem (4.4) in terms of the power diagrams $P_j$ associated to the points $\{y_j\}_{j=1}^n$ and the maximizer $\psi$ of $\mathcal{F}$.

**Theorem 4.2.** *Let $\mu \in \mathcal{P}_2(\mathbb{R}^d)$ be absolutely continuous with respect to the Lebesgue measure with Radon–Nikodym density $\rho(x)$. density $\rho(x)$. Let $\nu = \sum_{i=1}^n \nu_i \delta_{y_i}$. Let $\psi = (\psi_1, \cdots, \psi_n)$ be an maximizer of $\mathcal{F}$ defined in (4.7). Denote by $\{P_j\}_{j=1}^n$ the power diagrams associated to $\{y_j\}_{j=1}^n$ and $\psi$. Then the optimal transport plan $T$ solving the semi-discrete Monge's problem (4.4) is given by $T(x) = \nabla \bar{\varphi}(x)$, where $\bar{\varphi}(x) = \max_j \{x \cdot y_j + m_j\}$ for some $m_j \in \mathbb{R}$. Specifically, $T(x) = y_j$ if $x \in P_j(\psi)$.*

Theorem 4.2 shows that the optimal transport map in the semi-discrete case is achieved by the gradient of a particular piece-wise affine function that is the maximum of finitely many affine functions. A similar result was proved by [21] in the case where $\mu$ is defined on a compact convex domain. We provide a proof of Theorem 4.2, which deals with measures on the whole space $\mathbb{R}^d$ in Appendix E.2.

The next proposition shows that the piece-wise linear function $\max_j \{x \cdot y_j + m_j\}$ defined in Theorem 4.2 can be expressed exactly by a deep neural network.

**Proposition 4.1.** *Let $\bar{\varphi}(x) = \max_j\{x \cdot y_j + m_j\}$ with $\{y_j\}_{j=1}^n \subset \mathbb{R}^d$ and $\{m_j\}_{j=1}^n \subset \mathbb{R}$. Then there exists a fully connected deep neural network of depth $L = \lceil \log n \rceil$ and width $N = 2^L = 2^{\lceil \log n \rceil}$, and with ReLU activation function and parameters $\{W^\ell, b^\ell\}_{\ell=1}^{L+1}$ such that $\bar{\varphi} = \mathrm{DNN}(\{W^\ell, b^\ell\}_{\ell=1}^{L+1})$.*

The proof of Proposition 4.1 can be found in Appendix E.3. Theorem 4.1 is a direct consequence of Theorem 4.2 and Proposition 4.1.

## 5   Conclusion

In this paper, we establish that certain general classes of target distributions can be expressed arbitrarily well with respect to three type of IPMs by transporting a source distribution with maps which can be parametrized by DNNs. We provide upper bounds for the depths and widths of DNNs needed to achieve certain approximation error; the upper bounds are established with explicit dependence on the dimension of the underlying distributions and the approximation error.

## Broader Impact

This work focuses on theoretical properties of neural networks for expressing probability distributions. Our work can help understand the theoretical benefit and limitations of neural networks in approximating probability distributions under various integral probability metrics. Our work and the proof technique improve our understanding of the theoretical underpinnings of Generative Adversarial Networks and other generative models used in machine learning, and may lead to better use of these techniques with possible benefits to the society.

## Acknowledgments and Disclosure of Funding

The work of JL is supported in part by the National Science Foundation via grants DMS-2012286 and CCF-1934964 (Duke TRIPODS).

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
