[Supplementary Material]

# Supplementary Materials of "A Universal Approximation Theorem of Deep Neural Networks for Expressing Probability Distributions"

**Yulong Lu**
Department of Mathematics and Statistics
University of Massachusetts Amherst
Amherst, MA 01003
lu@math.umass.edu

**Jianfeng Lu**
Mathematics Department
Duke University
Durham, NC 27708
jianfeng@math.duke.edu

## A   Proof of Proposition 3.1

*Proof.* The proof follows from some previous results by [1] and [4]. In fact, in the one dimensional case, according to [1, Theorem 3.2], we know that if $\pi$ satisfies that

$$J_1(\pi) = \int_{-\infty}^{\infty} \sqrt{F(x)(1 - F(x))}dx < \infty \tag{A.1}$$

where $F$ is the cumulative distribution function of $\pi$, then for every $n \geq 1$,

$$\mathbf{E}\mathcal{W}_1(P_n, \pi) \leq \frac{J_1(\pi)}{\sqrt{n}}. \tag{A.2}$$

The condition (A.1) is fulfilled if $\pi$ has finite third moment since

$$J_1(\pi) \leq \int_0^{\infty} \sqrt{\mathbf{P}(|X| \geq x)}dx \leq 1 + \int_1^{\infty} \frac{\sqrt{\mathbf{E}|X|^3}}{x^{\frac{3}{2}}}dx = 1 + 2\sqrt{M_3}.$$

In the case that $d \geq 2$, it follows from that [4, Theorem 3.1] if $M_3 = \mathbb{E}_{X \sim \pi}|X|^3 d\pi < \infty$, then there exists a constant $c > 0$ independent of $d$ such that

$$\mathbf{E}\mathcal{W}_1(P_n, \pi) \leq cM_3^{1/3} \cdot \begin{cases} \frac{\log n}{\sqrt{n}} & \text{if } d = 2, \\ \frac{1}{n^{1/d}} & \text{if } d \geq 3. \end{cases} \tag{A.3}$$

$\square$

## B   Proof of Proposition 3.2

*Proof.* Thanks to [7, Proposition 3.1], one has that

$$\text{MMD}(P_n, \pi) = \left\| \int_{\mathbb{R}^d} k(\cdot, x)d(P_n - \pi)(x) \right\|_{\mathcal{H}_k}.$$

Let us define $\varphi(X_1, X_2, \cdots, X_n) := \|\int_{\mathbb{R}^d} k(\cdot, x)d(P_n - \pi)(x)\|_{\mathcal{H}_k}$. Then by definition $\varphi(X_1, X_2, \cdots, X_n)$ satisfies that for any $i \in \{1, \cdots, n\}$,

$$\left| \varphi(X_1, \cdots, X_{i-1}, X_i, \cdots, X_n) - \varphi(X_1, \cdots, X_{i-1}, X_i', \cdots, X_n) \right|$$
$$\leq \frac{2}{N} \sup_x \|k(\cdot, x)\|_{\mathcal{H}_k}$$
$$\leq \frac{2\sqrt{K_0}}{N}, \forall X_i, X_i' \in \mathbb{R}^d,$$

where we have used that $\|k(\cdot, x)\|_{\mathcal{H}_k} = \sup_x \sqrt{k(x,x)} \le \sqrt{K_0}$ by assumption. It follows from above and the McDiarmid's inequality that for every $\tau > 0$, with probability $1 - e^{-\tau}$,

$$\Big\| \int_{\mathbb{R}^d} k(\cdot, x) d(P_n - \pi)(x) \Big\|_{\mathcal{H}_k} \le \mathbf{E} \Big\| \int_{\mathbb{R}^d} k(\cdot, x) d(P_n - \pi)(x) \Big\|_{\mathcal{H}_k} + \sqrt{\frac{2\sqrt{K_0}\tau}{n}}.$$

In addition, we have by the standard symmetrization argument that

$$\mathbf{E} \Big\| \int_{\mathbb{R}^d} k(\cdot, x) d(P_n - \pi)(x) \Big\|_{\mathcal{H}_k} \le 2\mathbf{E}\mathbf{E}_\varepsilon \Big\| \frac{1}{n} \sum_{i=1}^n \varepsilon_i k(\cdot, X_i) \Big\|_{\mathcal{H}_k},$$

where $\{\varepsilon_i\}_{i=1}^n$ are i.i.d. Radmacher variables and $\mathbf{E}_\varepsilon$ represents the conditional expectation w.r.t $\{\varepsilon_i\}_{i=1}^n$ given $\{X_i\}_{i=1}^n$. To bound the right hand side above, we can apply McDiarmid's inequality again to obtain that with probability at least $1 - e^{-\tau}$,

$$\mathbf{E}\mathbf{E}_\varepsilon \Big\| \frac{1}{n} \sum_{i=1}^n \varepsilon_i k(\cdot, X_i) \Big\|_{\mathcal{H}_k} \le \mathbf{E}_\varepsilon \Big\| \frac{1}{n} \sum_{i=1}^n \varepsilon_i k(\cdot, X_i) \Big\|_{\mathcal{H}_k} + \sqrt{\frac{2\sqrt{K_0}\tau}{n}}$$

$$\le \Big( \mathbf{E}_\varepsilon \Big\| \frac{1}{n} \sum_{i=1}^n \varepsilon_i k(\cdot, X_i) \Big\|_{\mathcal{H}_k}^2 \Big)^{1/2} + \sqrt{\frac{2\sqrt{K_0}\tau}{n}}$$

$$\le \sqrt{\frac{\sqrt{K_0}}{n}} + \sqrt{\frac{2\sqrt{K_0}\tau}{n}},$$

where we have used Jensen's inequality for expectation in the second inequality and the independence of $\varepsilon_i$ and the definition of $K_0$ in the last inequality. Combining the estimates above yields that with probability at least $1 - 2e^{-\tau}$,

$$\mathrm{MMD}(P_n, \pi) = \Big\| \int_{\mathbb{R}^d} k(\cdot, x) d(P_n - \pi)(x) \Big\|_{\mathcal{H}_k} \le 2\sqrt{\frac{\sqrt{K_0}}{n}} + 3\sqrt{\frac{2\sqrt{K_0}\tau}{n}}.$$

$\square$

## C   Proof of Proposition 3.3

Thanks to [6, Theorem 3.6], $\mathrm{KSD}(P_n, \pi)$ is evaluated explicitly as

$$\mathrm{KSD}(P_n, \pi) = \sqrt{\mathbf{E}_{x,y \sim P_n}[u_\pi(x,y)]} = \sqrt{\frac{1}{n^2} \sum_{i,j=1}^n u_\pi(X_i, X_j)}, \tag{C.1}$$

where $u_\pi$ is a new kernel defined by

$$u_\pi(x,y) = s_\pi(x)^T k(x,y) s_\pi(y) + s_\pi(x)^T \nabla_y k(x,y)$$
$$+ s_\pi(y)^T \nabla_x k(x,y) + \mathrm{Tr}(\nabla_x \nabla_y k(x,y))$$

with $s_\pi(x) = \nabla \log \pi(x)$. Moreover, according to [6, Proposition 3.3], if $k$ satisfies Assumption K1, then $\mathrm{KSD}(P_n, \pi)$ is non-negative.

Our proof of Proposition 3.3 relies on the fact that $\mathrm{KSD}^2(P_n, \pi)$ can be viewed as a von Mises' statistics ($V$-statistics) and an important Bernstein type inequality due to [2] for the distribution of $V$-statistics, which gives a concentration bound of $\mathrm{KSD}^2(P_n, \pi)$ around its mean (which is zero). We recall this inequality in the theorem below, which is a restatement of [2, Theorem 1] for second order degenerate $V$-statistics.

### C.1   Bernstein type inequality for von Mises' statistics

Let $X_1, \cdots, X_n, \cdots$ be a sequence of i.i.d. random variables on $\mathbb{R}^d$. For a kernel $h(x,y) : \mathbb{R}^d \times \mathbb{R}^d \to \mathbb{R}$, we call

$$V_n = \sum_{i,j=1}^n h(X_i, X_j) \tag{C.2}$$

a von-Mises' statistic of order 2 with kernel $h$. We say that the kernel $h$ is *degenerate* if the following holds:

$$\mathbf{E}[h(X_1, X_2)|X_1] = \mathbf{E}[h(X_1, X_2)|X_2] = 0. \tag{C.3}$$

**Theorem C.1** ([2, Theorem 1]). *Consider the $V$-statistic $M_n$ defined by* (C.2) *with a degenerate kernel $h$. Assume the kernel satisfies that*

$$|h(x, y)| \leq g(x) \cdot g(y) \tag{C.4}$$

*for all $x, y \in \mathbb{R}^d$ with a function $g : \mathbb{R}^d \to \mathbb{R}$ satisfying for $\xi, J > 0$,*

$$\mathbf{E}[g(X_1)^k] \leq \xi^2 J^{k-2} k!/2, \tag{C.5}$$

*for all $k = 2, 3, \cdots$. Then there exist some generic constants $C_1, C_2 > 0$ independent of $k, h, l, \xi$ such that for any $t \geq 0$ that*

$$\mathbf{P}(|V_n| \geq n^2 t) \leq C_1 \exp\Big(-\frac{C_2 n t}{\xi^2 + J t^{1/2}}\Big). \tag{C.6}$$

**Remark C.1.** *As noted in [2, Remark 1], the inequality* (C.6) *is to some extent optimal. Moreover, a straightforward calculation shows that inequality* (C.6) *implies that for any $\delta \in (0, 1)$,*

$$\mathbf{P}\Big(\frac{1}{n^2}|V_n| \leq \frac{\mathscr{V}}{n}\Big) \geq 1 - \delta, \tag{C.7}$$

*where*

$$\mathscr{V} = \Big(\frac{J \log(\frac{C_1}{\delta})}{C_2} + \sqrt{\frac{\log(\frac{C_1}{\delta})}{C_2}} \xi\Big)^2.$$

## C.2 Moment bound of sub-Gaussian random vectors

Let us first recall a useful concentration result on sub-Gaussian random vectors.

**Theorem C.2** ([3, Theorem 2.1]). *Let $X \in \mathbb{R}^d$ be a sub-Gaussian random vector with parameters $m \in \mathbb{R}^d$ and $v > 0$. Then for any $t > 0$,*

$$\mathbf{P}\Big(|X - m| \geq v\sqrt{d + 2\sqrt{d}t + 2t}\Big) \leq e^{-t}. \tag{C.8}$$

*Moreover, for any $0 \leq \eta < \frac{1}{2v^2}$,*

$$\mathbf{E}\exp(\eta|X - m|^2) \leq \exp(v^2 d\eta + \frac{v^4 d\eta^2}{1 - 2v^2\eta}). \tag{C.9}$$

As a direct consequence of Theorem C.2, we have the following useful moment bound for sub-Gaussian random vectors.

**Proposition C.1.** *Let $X \in \mathbb{R}^d$ be a sub-Gaussian random vector with parameters $m \in \mathbb{R}^d$ and $v > 0$. Then for any $k \geq 2$,*

$$\mathbf{E}|X - m|^k \leq k\Big((2v\sqrt{d})^k + \frac{1}{2}(\frac{4v}{\sqrt{2}})^k k^{k/2}\Big). \tag{C.10}$$

*Proof.* From the concentration bound (C.8) and the simple fact that

$$d + 2\sqrt{d}t + 2t = 2\Big(\sqrt{t} + \frac{\sqrt{d}}{2}\Big)^2 + \frac{d}{2} \leq 4\Big(\sqrt{t} + \frac{\sqrt{d}}{2}\Big)^2,$$

one can obtain that

$$\mathbf{P}\Big(|X - m| \geq 2v\big(\sqrt{t} + \frac{\sqrt{d}}{2}\big)\Big) \leq \mathbf{P}\Big(|X - m| \geq v\sqrt{d + 2\sqrt{d}t + 2t}\Big) \leq e^{-t}.$$

Therefore, for any $s \geq v\sqrt{d}$, we obtain from above with $s = 2v(\sqrt{t} + \sqrt{d}/2)$ that

$$\mathbf{P}\Big(|X - m| \geq s\Big) \leq e^{-\big(\frac{s}{2v} - \frac{\sqrt{d}}{2}\big)^2}. \tag{C.11}$$

As a result, for any $k \geq 2$,

$$
\begin{aligned}
\mathbf{E}|X - m|^k &= \int_0^\infty \mathbf{P}(|X - m|^k \geq s)ds \\
&= \int_0^\infty \mathbf{P}(|X - m|^k \geq s^k)ks^{k-1}ds \\
&= \int_0^{2v\sqrt{d}} \mathbf{P}(|X - m| \geq s)ks^{k-1}ds + \int_{2v\sqrt{d}}^\infty \mathbf{P}(|X - m| \geq s)ks^{k-1}ds \\
&=: I_1 + I_2,
\end{aligned}
\tag{C.12}
$$

where we have used the change of variable $s \mapsto s^k$ in the second line above. It is clear that the first term

$$
I_1 \leq k(2v\sqrt{d})^k.
$$

For $I_2$, one first notices that if $s \geq 2v\sqrt{d}$, then $s/(2v) - \sqrt{d}/2 \geq s/(4v)$. Hence it follows from (C.8) that

$$
\begin{aligned}
I_2 &\leq \int_{2v\sqrt{d}}^\infty e^{-\left(\frac{s}{4v}\right)^2} ks^{k-1}ds \\
&= \frac{k(4v)^k}{2} \int_{\frac{d}{4}}^\infty e^{-t}t^{\frac{k-2}{2}}dt \\
&\leq \frac{k(4v)^k}{2}\Gamma(\frac{k}{2}).
\end{aligned}
$$

The last two estimates imply that

$$
\begin{aligned}
\mathbf{E}|X - m|^k &\leq k(2v\sqrt{d})^{k-1} + \frac{k(4v)^k}{2}\Gamma(\frac{k}{2}) \\
&\leq k\left((2v\sqrt{d})^k + \frac{1}{2}(\frac{4v}{\sqrt{2}})^k k^{k/2}\right),
\end{aligned}
$$

where the second inequality above follows from $\Gamma(\frac{k}{2}) \leq (k/2)^{k/2}$ for $k \geq 2$. □

### C.3 Proof of Proposition 3.3

Our goal is to invoke Theorem C.1 to obtain a concentration inequality for KSD. Recall that $\mathrm{KSD}(P_n, \pi)$ is defined by

$$
\mathrm{KSD}^2(P_n, \pi) = \frac{1}{n^2}\sum_{i,j=1}^n u_\pi(X_i, X_j)
$$

with the kernel

$$
u_\pi(x, y) = s_\pi(x)^T k(x, y)s_\pi(y) + s_q(x)^T\nabla_y k(x, y) + s_q(y)^T\nabla_x k(x, y) + \mathrm{Tr}(\nabla_x \nabla_y k(x, y)).
$$

Let us first verify that the new kernel $u_\pi$ satisfies the assumption of Theorem C.1. In fact, since $s_\pi(x) = \nabla \log(\pi(x))$, one obtains from integration by part that

$$
\begin{aligned}
\mathbf{E}[u_\pi(X_1, X_2)|X_1 = x] &= \int_{\mathbb{R}^d} u_\pi(x, y)d\pi(y) \\
&= \int_{\mathbb{R}^d} s_\pi(x)k(x, y)s_\pi(y) + s_q(x)^T\nabla_y k(x, y) \\
&\quad + s_\pi(y)^T\nabla_x k(x, y) + \mathrm{Tr}(\nabla_x \nabla_y k(x, y))d\pi(y) \\
&= \int_{\mathbb{R}^d} k(x, y)s_\pi(x)^T\nabla_y \pi(y)dy - \int_{\mathbb{R}^d} \nabla_y \cdot (s_\pi(x)\pi(y))dy \\
&\quad + \int_{\mathbb{R}^d} \nabla_y \pi(y)^T\nabla_x k(x, y)dy - \int_{\mathbb{R}^d} \nabla_y \pi(y)^T\nabla_x k(x, y)dy \\
&= 0.
\end{aligned}
$$

Similarly, one has
$$\mathbf{E}[u_\pi(X_1, X_2)|X_2 = y] = 0.$$
This shows that $u_\pi$ satisfies the condition of degeneracy (C.3).

Next, we show that $u_\pi$ satisfies the bound (C.4) with a function $g$ satisfying the moment condition (C.5). In fact, by Assumption K3 on the kernel $k$ and Assumption 1 on the target density $\pi$,

$$
\begin{aligned}
|u_\pi(x,y)| &\leq L^2 K_1(1+|x|) \cdot (1+|y|) + LK_1(1+|x|+1+|y|) + K_1(1+d) \\
&\leq K_1(L+1)^2(\sqrt{d}+1+|x|) \cdot (\sqrt{d}+1+|y|) \\
&=: g(x) \cdot g(y),
\end{aligned}
$$

where $g(x) = \sqrt{K_1}(L+1)(\sqrt{d}+1+|x|)$ and the constant $L$ is defined in (2.6). To verify $g$ satisfies (C.5), we write

$$
\begin{aligned}
\mathbf{E}_{X\sim\pi}[g(X)^k] &= (\sqrt{K_1}(L+1))^k \mathbf{E}_{X\sim\pi}[(\sqrt{d}+1+|X|)^k] \\
&\leq (\sqrt{K_1}(L+1))^k \mathbf{E}_{X\sim\pi}[(\sqrt{d}+1+|m|+|X-m|)^k] \\
&= (\sqrt{K_1}(L+1))^k \left((\sqrt{d}+1+|m|)^k + \sum_{j=1}^{k}\binom{k}{j}(\sqrt{d}+1+|m|)^{k-j}\mathbf{E}_{X\sim\pi}|X-m|^j\right).
\end{aligned}
\tag{C.13}
$$

Thanks to Proposition C.1, we have for any $j \geq 1$,

$$
\begin{aligned}
\mathbf{E}_{X\sim\pi}|X-m|^j &\leq \left(\mathbf{E}_{X\sim\pi}|X-m|^{2j}\right)^{1/2} \\
&\leq (2j)^{1/2} \cdot \left((2\upsilon\sqrt{d})^{2j} + \frac{1}{2}\left(\frac{4\upsilon}{\sqrt{2}}\right)^{2j} \cdot (2j)^j\right)^{1/2} \\
&\leq (2j)^{1/2} \cdot \left(2\max(4\upsilon, 1) \cdot \sqrt{d} \cdot \sqrt{j}\right)^j \\
&\leq \left(2e^{1/e}\max(4\upsilon, 1) \cdot \sqrt{d} \cdot \sqrt{j}\right)^j,
\end{aligned}
\tag{C.14}
$$

where we have used the simple fact that $(2j)^{1/(2j)} \leq e^{1/e}$ for any $j \geq 1$ in the last inequality. Plugging (C.14) into (C.13) yields that

$$
\begin{aligned}
\mathbf{E}_{X\sim\pi}[g(X)^k] &\leq 2(\sqrt{K_1}(L+1))^k \left(\sqrt{d}+1+|m|+2e^{1/e}\max(4\upsilon, 1)\sqrt{d} \cdot \sqrt{k}\right)^k \\
&= 2(\sqrt{K_1}(L+1))^k \exp\left(k\log\left(\sqrt{d}+1+|m|+2e^{1/e}\max(4\upsilon, 1)\sqrt{d} \cdot \sqrt{k}\right)\right).
\end{aligned}
\tag{C.15}
$$

Using the fact that $\log(a+b) - \log(a) = \log(1+b/a) \leq b/a$ for all $a, b \geq 1$, one has

$$
\begin{aligned}
&\exp\left(k\log\left(\sqrt{d}+1+|m|+2e^{1/e}\max(4\upsilon, 1)\sqrt{d} \cdot \sqrt{k}\right)\right) \\
&\leq \exp\left(k\log\left(\underbrace{2e^{1/e}\max(4\upsilon, 1)\sqrt{d} \cdot \sqrt{k}}_{=:A}\right)\right) \cdot \exp\left(\sqrt{k} \cdot \underbrace{\frac{\sqrt{d}+1+|m|}{2e^{1/e}\max(4\upsilon, 1)\sqrt{d}}}_{=:B}\right).
\end{aligned}
\tag{C.16}
$$

Since by assumption $|m| \leq m^*\sqrt{d}$ and $d \geq 1$, we have

$$B \leq \frac{2+m^*}{2e^{1/e}\max(4\upsilon, 1)} =: \tilde{B}.$$

As a consequence of above and the fact that $k! \geq \left(\frac{k}{3}\right)^k$ for any $k \in \mathbb{N}_+$,

$$
\begin{aligned}
\exp\left(k\log\left(\sqrt{d}+1+|m|+2e^{1/e}\max(4\upsilon, 1)\sqrt{d} \cdot \sqrt{k}\right)\right) &\\
\leq \exp(k\log k/2) \cdot \exp(k(\log A + \tilde{B})) &\\
= \left(\frac{k}{3}\right)^{k/2} \cdot \left(\sqrt{3}A\exp(\tilde{B} + \frac{1}{2})\right)^k &\\
\leq k! \cdot \left(\sqrt{3}A\exp(\tilde{B} + \frac{1}{2})\right)^k. &
\end{aligned}
\tag{C.17}
$$

Combining this with (C.15) implies that the moment bound assumption (C.5) holds with the constants

$$J = \sqrt{3K_1}(L+1)A\exp\left(\tilde{B} + \frac{1}{2}\right) \text{ and } \xi = 2J.$$

Therefore it follows from the definition of $\mathrm{KSD}(P_n, \pi)$ in (C.1) and the concentration bound (C.7) implied by Theorem C.1 that with at least probability $1 - \delta$,

$$\mathrm{KSD}(P_n, \pi) \leq \frac{C}{\sqrt{n}},$$

with the constant

$$C = J\Big(\frac{\log(\frac{C_1}{\delta})}{C_2} + 2\sqrt{\frac{\log(\frac{C_1}{\delta})}{C_2}}\Big).$$

Since by definition the constant $A = \mathcal{O}(\sqrt{d})$ for large $d$, we have that the constant $C = \mathcal{O}(\sqrt{d})$. This completes the proof.

# D   Summarizing Propositions 3.1 - 3.3

The theorem below summarizes the Propositions 3.1 - 3.3 above, serving as one of the ingredients for proving Theorem 2.1.

**Theorem D.1.** *Let $\pi$ be a probability measure on $\mathbb{R}^d$ and let $P_n = \frac{1}{n}\sum_{i=1}^{n}\delta_{X_i}$ be the empirical measure associated to the i.i.d. samples $\{X_i\}_{i=1}^n$ drawn from $\pi$. Then we have the following:*

1. *If $\pi$ satisfies $M_3 = \mathbf{E}_{X\sim\pi}|X|^3 < \infty$, then there exists a realization of empirical measure $P_n$ such that*

$$\mathcal{W}_1(P_n, \pi) \leq C \cdot \begin{cases} n^{-1/2}, & d = 1, \\ n^{-1/2}\log n, & d = 2, \\ n^{-1/d}, & d \geq 3, \end{cases}$$

   *where the constant $C$ depends only on $M_3$.*

2. *If $k$ satisfies Assumption K2 with constant $K_0$, then there exists a realization of empirical measure $P_n$ such that*

$$\mathrm{MMD}(P_n, \pi) \leq \frac{C}{\sqrt{n}},$$

   *where the constant $C$ depending only on $K_0$.*

3. *If $\pi$ satisfies Assumption 1 and 2 and $k$ satisfies Assumption K3 with constant $K_1$, then there exists a realization of empirical measure $P_n$ such that*

$$KSD\,(P_n, \pi) \leq C\sqrt{\frac{d}{n}},$$

   *where the constant $C$ depends only on $L, K_1, m^*, \upsilon$.*

# E   Semi-discrete optimal transport with quadratic cost

## E.1   Structure theorem of optimal transport map

We recall the structure theorem of optimal transport map between $\mu$ and $\nu$ under the assumption that $\mu$ does not give mass to null sets.

**Theorem E.1** ([8, Theorem 2.9 and Theorem 2.12]). *Let $\mu$ and $\nu$ be two probability measures on $\mathbb{R}^d$ with finite second moments. Assume that $\mu$ is absolutely continuous with respect to the Lebesgue measure. Consider the functionals $\mathcal{K}$ and $\mathcal{J}$ defined in Monge's problem (4.1) and dual Kantorovich problem (4.2) with $c = \frac{1}{2}|x - y|^2$. Then*

*(i) there exists a unique solution $\pi$ to Kantorovich's problem, which is given by $\pi(dxdy) = (\mathbf{Id} \times T)_{\#}\mu$ where $T(x) = \nabla\bar{\varphi}(x)$ $\mu$-a.e.x for some convex function $\bar{\varphi} : \mathbb{R}^d{\to}\mathbb{R}$. In another word, $T(x) = \nabla\bar{\varphi}(x)$ is the unique solution to Monge's problem.*

*(ii) there exists an optimal pair $(\varphi(x), \varphi^c(y))$ or $(\psi^c(x), \psi(y))$ solving the dual Kantorovich's problem, i.e. $\sup_{(\varphi,\psi)\in\Phi_c} \mathcal{J}(\varphi, \psi) = \mathcal{J}(\varphi, \varphi^c) = \mathcal{J}(\psi^c, \psi)$;*

*(iii) the function $\bar{\varphi}(x)$ can be chosen as $\bar{\varphi}(x) = \frac{1}{2}|x|^2 - \varphi(x)$ (or $\bar{\varphi}(x) = \frac{1}{2}|x|^2 - \psi^c(x)$) where $(\varphi(x), \varphi^c(y))$ (or $(\psi^c(x), \psi(y))$) is an optimal pair which maximizes $\mathcal{J}$ within the set $\Phi_c$.*

## E.2 Proof of Theorem 4.2

Recall that the dual Kantorovich problem in the semi-discrete case reduces to maximizing the following functional

$$\mathcal{F}(\psi) = \int \inf_j \left( \frac{1}{2}|x - y_j|^2 - \psi_j \right) \rho(x)dx + \sum_{j=1}^n \psi_j \nu_j. \tag{E.1}$$

Proof of Theorem 4.2 relies on two useful lemmas on the functional $\mathcal{F}$. The first lemma below shows that the functional $\mathcal{F}$ is concave, whose proof adapts that of [5, Theorem 2] for semi-discrete optimal transport with the quadratic cost.

**Lemma E.1.** *Let $\rho$ be a probability density on $\mathbb{R}^d$. Let $\{y_j\}_{j=1}^n \subset \mathbb{R}^d$ and let $\{\nu_j\}_{j=1}^n \subset [0,1]$ be such that $\sum_{j=1}^n \nu_j = 1$. Then the functional $\mathcal{F}$ be defined by (E.1) is concave.*

*Proof.* Let $\mathcal{A}: \mathbb{R}^d \to \{1, 2, \cdots, n\}$ be an assignment function which assigns a point $x \in \mathbb{R}^d$ to the index $j$ of some point $y_j$. Let us also define the function

$$\widetilde{\mathcal{F}}(\mathcal{A}, \psi) = \int \left( \frac{1}{2}|x - y_{\mathcal{A}(x)}|^2 - \psi_{\mathcal{A}(x)} \right)\rho(x)dx + \sum_{j=1}^n \psi_j \nu_j.$$

Then by definition $\mathcal{F}(\psi) = \inf_{\mathcal{A}} \widetilde{\mathcal{F}}(\mathcal{A}, \psi)$. Denote $\mathcal{A}^{-1}(j) = \{x \in \mathbb{R}^d | \mathcal{A}(x) = j\}$. Then

$$\widetilde{\mathcal{F}}(\mathcal{A}, \psi) = \sum_{j=1}^n \left[ \int_{\mathcal{A}^{-1}(j)} \left( \frac{1}{2}|x - y_j|^2 - \psi_j \right)\rho(x)dx + \psi_j \nu_j \right]$$

$$= \sum_{j=1}^n \int_{\mathcal{A}^{-1}(j)} \frac{1}{2}|x - y_j|^2 \rho(x)dx + \sum_{j=1}^n \psi_j \left( \nu_j - \int_{\mathcal{A}^{-1}(j)} \rho(x)dx \right).$$

Since the function $\widetilde{\mathcal{F}}(\mathcal{A}, \psi)$ is affine in $\psi$ for every $\mathcal{A}$, it follows that $\mathcal{F}(\psi) = \inf_{\mathcal{A}} \widetilde{\mathcal{F}}(\mathcal{A}, \psi)$ is concave. $\square$

The next lemma computes the gradient of the concave function $\mathcal{F}$; see [5, Section 7.4] for the corresponding result with general transportation cost.

**Lemma E.2.** *Let $\rho$ be a probability density on $\mathbb{R}^d$. Let $\{y_j\}_{j=1}^n \subset \mathbb{R}^d$ and let $\{\nu_j\}_{j=1}^n \subset [0,1]$ be such that $\sum_{j=1}^n \nu_j = 1$. Denote by $P_j(\psi)$ the power diagram associated to $\psi$ and $y_j$. Then*

$$\partial_{\psi_i} \mathcal{F}(\psi) = \nu_i - \mu(P_i(\psi)) = \nu_i - \int_{P_i(\psi)} \rho(x)dx. \tag{E.2}$$

*Proof.* By the definition of $\mathcal{F}$ in (E.1), we rewrite $\mathcal{F}$ as

$$\mathcal{F}(\psi) = \int \frac{1}{2}|x|^2 \rho(dx) + \int \inf_j \left\{ -x \cdot y_j + \frac{1}{2}|y_j|^2 - \psi_j \right\}\rho(x)dx + \sum_{j=1}^n \psi_j \nu_j$$

$$= \int \frac{1}{2}|x|^2 \rho(dx) - \int \sup_j \left\{ x \cdot y_j + \psi_j - \frac{1}{2}|y_j|^2 \right\}\rho(x)dx + \sum_{j=1}^n \psi_j \nu_j$$

To prove (E.2), it suffices to prove that

$$\partial_{\psi_i} \left( \int \sup_j \left\{ x \cdot y_j + \psi_j - \frac{1}{2}|y_j|^2 \right\}\rho(x)dx \right) = \int_{P_i(\psi)} \rho(x)dx. \tag{E.3}$$

Note that the partial derivative on the left side of above makes sense since $g(x, \psi) := \sup_j \{x \cdot y_j + \psi_j - \frac{1}{2}|y_j|^2\}$ is convex with respect to $(x, \psi)$ on $\mathbb{R}^d \times \mathbb{R}^d$ so that the resulting integral against the measure $\rho$ is also convex (and hence Lipschitz) in $\psi$. To see (E.3), since $g(x, \psi)$ is convex and piecewise linear in $\psi$ for any fixed x, it is easy to observe that

$$\partial_{\psi_i} g(x, \psi) = \delta_{ij} \text{ if } x \in \left\{ x \in \mathbb{R}^d \Big| x \cdot y_j + \psi_j - \frac{1}{2}|y_j|^2 = g(x, \psi) \right\}.$$

However, by subtracting $\frac{1}{2}|x|^2$ on both sides of the equation inside the big parenthesis and then flipping the sign one sees that

$$\left\{ x \in \mathbb{R}^d \Big| x \cdot y_j + \psi_j - \frac{1}{2}|y_j|^2 = g(x, \psi) \right\} = P_j(\psi).$$

Namely we have obtained that

$$\partial_{\psi_i} g(x, \psi) = \delta_{i,j} \text{ if } x \in P_j(\psi).$$

In particular, this implies that $\psi \to g(x, \psi)$ is 1-Lipschitz in $\psi$ uniformly with respect to $x$. Finally since $\rho(x)$ is a probability measure, the desired identity (E.2) follows from the equation above and the dominated convergence theorem. This completes the proof of the lemma. $\square$

With the lemmas above, we are ready to prove Theorem 4.2. In fact, according to Lemma E.1 and Lemma E.2, $\psi = (\psi_1, \cdots, \psi_n)$ is a maximizer of the functional $\mathcal{F}$ if and only if

$$\partial_{\psi_i} \mathcal{F}(\psi) = \nu_i - \mu(P_i(\psi)) = \nu_i - \int_{P_i(\psi)} \rho(x)dx = 0.$$

Since the dual Kantorovich problem in the semi-discrete setting reduces to the problem of maximizing $\mathcal{F}$, it follows from Theorem E.1 that the optimal transport map $T$ solving the semi-discrete Monge's problem (4.4) is given by $T(x) = \nabla \bar{\varphi}(x)$ where $\bar{\varphi}(x) = \frac{1}{2}|x|^2 - \varphi(x)$ and $\varphi(x) = \min_j \frac{1}{2}|x - y_j|^2 - \psi_j$. Consequently,

$$\begin{aligned}
\bar{\varphi}(x) &= \frac{1}{2}|x|^2 - \varphi(x) \\
&= \frac{1}{2}|x|^2 - \left( \min_j \{ \frac{1}{2}|x - y_j|^2 - \psi_j \} \right) \\
&= \max_j \{x \cdot y_j + m_j\}
\end{aligned}$$

with $m_j = \psi_j - \frac{1}{2}|y_j|^2$. Moreover, noticing that $\varphi(x)$ can be rewritten as

$$\varphi(x) = \frac{1}{2}|x - y_j|^2 - \psi_j \text{ if } x \in P_j(\psi),$$

one obtains that $T(x) = \nabla \bar{\varphi}(x) = y_j$ if $x \in P_j(\psi)$.

### E.3   Proof of Proposition 4.1

Let us first consider the case that $n = 2^k$ for some $k \in \mathbb{N}$. Then

$$\bar{\varphi}(x) = \max_{j=1,\cdots,2^k} \{x \cdot y_j + m_j\} = \max_{j=1,\cdots,2^{k-1}} \max_{i \in \{2j-1,2j\}} \{x \cdot y_i + m_i\}.$$

Let us define maps $\varphi_n : \mathbb{R}^n \to \mathbb{R}^{n/2}$ and $\psi : \mathbb{R}^d \to \mathbb{R}^n$ by setting

$$[\varphi_n(z)]_i = \max\{z_{2i-1}, z_{2i}\}, i = 1, \cdots, n/2 \text{ and } [\psi(x)]_j = x \cdot y_j + m_j, j = 1, \cdots, n.$$

Then by definition it is straightforward that

$$\bar{\varphi}(x) = (\varphi_2 \circ \varphi_4 \circ \cdots \circ \varphi_{n/2} \circ \varphi_n \circ \psi)(x). \tag{E.4}$$

By defining

$$Y = \begin{pmatrix} y_1^T \\ y_2^T \\ \vdots \\ y_n^T \end{pmatrix}, m = \begin{pmatrix} m_1 \\ m_2 \\ \vdots \\ m_n \end{pmatrix},$$

we can write the map $\psi$ as
$$\psi(x) = Y \cdot x + m. \tag{E.5}$$
Moreover, thanks to the following simple equivalent formulation of the maximum function:
$$\max(a, b) = \text{ReLU}(a - b) + \text{ReLU}(b) - \text{ReLU}(-b)$$
$$= h^T \cdot \text{ReLU}\left(A \begin{pmatrix} a \\ b \end{pmatrix}\right),$$
where
$$A = \begin{pmatrix} 1 & -1 \\ 0 & 1 \\ 0 & -1 \end{pmatrix}, \quad h = \begin{pmatrix} 1 \\ 1 \\ -1 \end{pmatrix},$$
we can express the map $\varphi_n$ in terms of a two-layer neural network as follows
$$\varphi_n(z) = H_{n/2} \cdot \text{ReLU}(A_n \cdot z), \tag{E.6}$$
where $A_n = \oplus^n A$ and $H_n = \oplus^n h$. Finally, by combining (E.4), (E.5) and (E.6), one sees that $\bar{\varphi}$ can be expressed in terms of a DNN of width $n$ and depth $\log n$ with parameters $(W^\ell, b^\ell)_{\ell=1}^{L+1}$ defined by
$$W^0 = A_n \cdot Y, \quad b^0 = A_n \cdot m,$$
$$W^1 = A_{n/2} \cdot H_{n/2}, \quad b^1 = 0,$$
$$W^2 = A_{n/4} \cdot H_{n/4}, \quad b^2 = 0,$$
$$\cdots,$$
$$W_{L-1} = A_2 \cdot H_2, \quad b^{L-1} = 0,$$
$$W_L = H_1, \quad b^L = 0.$$
In the general case where $\log_2 n \notin \mathbb{N}$, we set $k = \lceil \log_2 n \rceil$ so that $k$ is smallest integer such that $2^k > n$. By redefining $y_j = 0$ and $m_j = 0$ for $j = n + 1, \cdots, 2^k$, we may still write $\bar{\varphi}(x) = \max_{j=1,\cdots,2^k}\{x \cdot y_j + m_j\}$ so that the analysis above directly applies.

## F  Proof of Main Theorem 2.1

The proof follows directly from Theorem D.1 and Theorem 4.1. Indeed, on the one hand, the quantitative estimate for the convergence of the empirical measure $P_n$ directly translates to the sample complexity bounds in Theorem 2.1 with a given error $\varepsilon$. On the other hand, Theorem 4.1 provides a push-forward from $p_x$ to the empirical measure $P_n$ based on the gradient of a DNN.