[Reviews · NeurIPS 2020]

Review 1

Summary and Contributions: This paper examines the ability to approximate a distribution constructed by push-forward with a neural network. As generative models, flow models, etc. become more popular, it is very common to construct new distributions by extruding distributions with generators. This paper proves that when a generator is constructed with a neural network, a constant distribution can be approximated with a sufficiently large network and under arbitrary errors. Errors on the distribution were measured in Wasserstein, MMD, and KSD, and the rate of width and breadth required for the error sought was also revealed.

Strengths: This is an important result that neatly formulates a widely treated problem. The results are also good and general.

Weaknesses: I see this paper as a positive, but I have the following unclear points. Is it possible to describe the number of weights needed for the network for the approximation? Some important approximation capability papers investigates a relation btw a number of their weights and the approximation power. How does this affect them? Is it possible to give a similar rate if there is no density in the distribution? GANs, for example, generates a distribution in high-dimensional space by push-forward a low-dimensional noise distribution. Therefore, the generated distribution does not possess a natural density function. What would be the result in that case? Regarding citations, I felt that some relevant papers were missing. In the context of the flow model, there are similar studies that should discuss the differences. The last paper (Lin+ 2018) is the universality of ResNet, but since it is for an R^d to R^d mapping, it can be discussed as a generator. Kong, Z., & Chaudhuri, K. (2020). The Expressive Power of a Class of Normalizing Flow Models. arXiv preprint arXiv:2006.00392. Bailey, B., & Telgarsky, M. J. (2018). Size-noise tradeoffs in generative networks. In Advances in Neural Information Processing Systems (pp. 6489-6499). Lin, H., & Jegelka, S. (2018). Resnet with one-neuron hidden layers is a universal approximator. In Advances in neural information processing systems (pp. 6169-6178).

Correctness: Good enough, as far as I've checked.

Clarity: It's written clearly enough.

Relation to Prior Work: A few papers are missing references, but they are not fatal.

Reproducibility: Yes

Additional Feedback:


Review 2

Summary and Contributions: This paper proves a universal approximation theorem for probability distributions, instead of continuous functions. The authors build an intuitive proof based on first using empirical measures. The authors also show that these empirical measures converge to the correct target measures as more data is collected. These convergence rates are calculated for the 1-Wasserstein distance, MMD, and KSD.

Strengths: The math is very precise and well presented. The authors did an excellent job with presenting the complexities of their theory in a very clear way. Line 173 - The high level comparison between the authors’ work and the proof of the universal approximation theorem is very clear and intuitive. These types of theorems are usually hard to digest and this paragraph is very helpful for the reader to read over before jumping into the details. The structure of the paper is very easy to follow and clear. The authors present their results excellently.

Weaknesses: Line 62 - The authors’ third contribution doesn’t sound like a contribution, honestly. Constructing an explicit neural network is novel. Of course, it helps that the authors are not talking about arbitrary functions, which are hard to relate to neural networks. The authors present several results from other papers in their work. This is fine, but it seems like too much of this work is just referencing other papers. For example, the authors make it sound like Proposition 1 and 2 are trivial or known results. Moreover, they make it seem like Lines 248-274 are a review from other papers. This is not inherently bad because it helps the reader understand more about the theorems that the authors prove, but it reduces the novel contributions that the authors are able to present in an 8-page paper.

Correctness: Line 130 - I think that the authors want a strict inequality on Assumption K1 because they mention that the measures are non-zero.

Clarity: The paper is extremely clear and well written. Bravo. Below are a few suggestions on parts that I was confused on. There are a few small typos. Line 72 - network network should probably be neural network. Line 185 - rational should be rationale. Weird space on Line 218. Equation 4.2 - gamma(dxdy) should be gamma(dx,dy). Line 286 - linear should be affine. Line 113 - The equation for Wasserstein distance, MMD, and KSD all use inputs p and pi but Equation 2.2 uses p and q. Mathematically this is correct but is not as consistent as it could be and causes confusion at first glance because the notation switched. Line 133 - Is the maximum taken over all m, n > 0 so that m + n <= 1? If so, then I don’t understand why k needs to be twice differentiable, instead of just once differentiable. Because the two Equations in (2.5) only need k to be once differentiable. Theorem 4.1 - The authors did not define \mathcal P_2(R^d) yet. At first glance, this may seem like the space of degree 2 polynomials defined on R^d, so I looked in the appendix to find out they mean that mu has finite second moments. The authors should mention this somewhere in the main text for the reader. Moreover, by “Lebesgue density” I think that the authors mean that mu is an absolutely continuous measure, but it is not clear.

Relation to Prior Work: Prior work is thoroughly discussed and well presented.

Reproducibility: Yes

Additional Feedback: Overall, I really enjoyed this paper. The authors did an excellent job.


Review 3

Summary and Contributions: This paper proves that the gradients of deep ReLU networks can map a given probability to a sequence of probability density distributions converging to a target distribution measured by Wasserstein distance, MMD or KSD of a certain kernels. This is the first work obtaining a sufficiently general result on the approximation power of neural networks for probability distributions.

Strengths: The statement is clear and solid in theory. It extends the approximation theory of neural networks to the regime of probability distributions, and adds our knowledge on generative models.

Weaknesses: 1. This connection between the optimal transportation theory and generative models have been shown by previous literatures, for example Ref 30 given in the paper. The proofs in this work are simple applications of the well-known optimal transportation theory and the results are quite incremental and not surprising. 2. People will care more about the approximation capability of the models proven to be useful in practice. This paper only makes a statement on the gradient networks of deep ReLU networks. It is not clear how to train such networks in practice or how they are related to popular probability modeling methods like GAN, VAE or normalizing flow.

Correctness: I think the claims are correct, though I do not carefully check all the details of the proof.

Clarity: The paper is easy to read and statements are clear.

Relation to Prior Work: It describes its difference from previous theoretical work. But adding more words on the implication of this result for algorithms designed from the same optimal transport perspective will be better.

Reproducibility: Yes

Additional Feedback:

[Author Response · NeurIPS 2020]

We are grateful to the referees for positive evaluations and useful suggestions for our work. Below we respond to the
comments by each referee.

R1: It is possible to formulate our approximation result in terms of the number of weights. In fact, our construction
of network is explicit and has number of weights = $N^2(L+1)$ where $N$ is the width and $L$ is the depth of the neural
networks. This combines with Theorem 2.1 of the paper would lead to an estimate of number of weights needed to
achieve certain approximation error. We will add a remark on this in the revised version. Thanks for the suggestion.

This paper only studies the approximation power of neural nets for generating distributions with an identical dimension.
It is certainly very important and interesting to consider the dimension mismatch case, particularly the case where
the dimension of the input distribution is much smaller than that of the output. However, constructing neural network
based transport maps between generic distributions with unequal dimensions is highly non-trivial. Our result relies
strongly on Brenier's theory of OT maps which is valid only in the equal-dimensional case. OT theory between unequal
dimensions is rather challenging and characterizing the OT map in such case is far less understood. We will study this
problem and report the results in future work.

Thanks very much for pointing out the relevant references. Let us briefly compare the results in those references
with ours. Kong and Chaudhuri analyzed the expressive power of normalizing flow models under the $L^1$-norm of
distributions and showed that the flow models have limited approximation capability in high dimensions. We show that
feedforward DNNs can approximate a general class of distributions in high dimensions with respect to three IPMs. Lin
and Jegelka studied the universal approximation of certain ResNets for multivariate functions; the approximation result
there however was not quantitative and did not consider the universal approximation of ResNets for distributions. The
work by Bailey and Telgarsky is closest to ours, but they only studied the approximation power of DNNs for expressing
uniform and Gaussian distributions in the Wasserstein distance, whereas we proved quantitative approximation results
for fairly general distributions under three IPMs. We will cite and comment on the references in the revised version.

R2: We listed the third point of the contribution in the paper to emphasize that the transport map we would like to
approximate is highly structured, namely the maximum of finitely many affine functions. Thanks to the simple structure,
the parameters of the ReLU DNN can be explicitly determined. We understand the referee's concern on this point as a
contribution, and we will change this point as a remark on our results instead of a contribution in the revision.

We agree with the referee that Proposition 3.1 and Proposition 3.2 are known results. However, we state these results
in a way that the dependencies of the convergence estimates on some key parameters (e.g. dimensionality) are more
explicit than that in the literature. This is required for us to obtain explicit error estimates.

We understand that the referee feels that lines 248-274 about optimal transport might take too much space to be presented
in an 8-page paper. We think however they are necessary for at least two reasons. First, since we are using optimal
transport (with quadratic cost) to build the transport map between distributions, it is essential to recall the set-up of the
optimal transport problem and the dual formulation, and also beneficial for readers without background knowledge
on optimal transport. In addition, to describe the formulation of the optimal transport map in the semi-discrete case
in Theorem 4.2, we need to introduce some notations such as the dual variable $\psi$ and the power diagrams $P_j$. We
also emphasize that a version of Theorem 4.2 for target measure defined on a compact convex domain was proved by
Ref [20]; that result was however not enough for our purpose as the measures in our work are defined on the whole
space $\mathbb{R}^d$. As a result, we come up with a different proof strategy based on purely the duality argument, instead of the
geometric arguments used in Ref [20].

About Assumption K1 – Yes, the inequality should be strict. Thanks for catching this.

Thanks for pointing out the typos. We will correct them in the revised version. We will use $D(p, \pi)$ instead of $D(p, q)$
to avoid notation inconsistency. We require $k(x, y)$ to be twice differentiable since in the second condition of equation
(2.5) we differentiate $k$ with respect to both $x$ and $y$. We will add the definition of $\mathcal{P}_2(\mathbb{R}^d)$ in the notation part of Section
1. We will use "absolute continuous with respect to Lebesgue" instead of "Lebesgue density" in the revision.

R3: We agree with the referee that the connection between optimal transport and generative models have been explored
in the literature. To the best of our knowledge, however, this work is the first proof of a general and quantified universal
approximation result of DNNs for distributions by using the optimal transport theory.

This work focuses mainly on the theoretical expressibility of DNNs for representing probability distributions. The
transport map constructed in the paper is parameterized by the gradient of a DNN instead of a DNN itself and this
gradient formulation originates essentially from the Brenier's theory for the optimal transport map. A discussion of the
gradient formulation for practical training was included in the last paragraph of Section 2 and we plan to further expand
that in the revision. Although the gradient parametrization is not as common as parametrizing the map itself, there have
been an increasing number of works utilizing the gradients of neural networks for various practical leaning problems
including GANs; see the discussion and the references mentioned at the beginning of page 6. Thanks for the comments.

[Meta-Review · NeurIPS 2020]

This paper shows that the gradients of certain ResNets can serve as generators to produce any of a broad class of distributions, measuring quality in several different metrics, including empirical measures. Pushing forward the gradient of a network rather than the network itself is somewhat unusual, and the paper requires a latent dimension the same size as the ambient dimension of the target distribution. Nevertheless, the proof is satisfying, explicit, and clear. This paper makes a nice contribution to the theory of generative models.